# Vitamin B3 Ameliorates Sleep Duration and Quality in Clinical and Pre-Clinical Studies

**DOI:** 10.3390/nu17121982

**Published:** 2025-06-12

**Authors:** Carleara Weiss

**Affiliations:** School of Nursing, University at Buffalo, The State University of New York, New York, NY 14214, USA; carleara@buffalo.edu

**Keywords:** vitamin B3, nicotinamide riboside, sleep

## Abstract

NAD^+^ is a fundamental molecule participating as a redox cofactor in several metabolic reactions and has a neuroprotective role associated with oxidate stress. Despite its critical role, NAD^+^ levels sharply decline with age, contributing to the pathogenesis of aging-related diseases. Supplementation with nicotinamide riboside (NR), also known as a form of vitamin B3, a biochemical precursor of NAD^+^, may replenish this depletion. **Background/Objectives**: Mounting evidence suggests that dietary supplementation with NR, a form of vitamin B3 and a biochemical precursor of NAD^+^, enhances NAD^+^ bioavailability and prevents the detrimental effects on sleep, cognitive function, mitochondrial function, and insulin sensitivity. However, there is a paucity of studies focused on how NR administration affects sleep patterns. This narrative review summarizes the current state of scientific knowledge on the effects of nicotinamide riboside supplementation on sleep. **Results**: Pre-clinical studies indicate that NR enhances the performance of the clock genes BMAL1 and PER2, and ameliorates chronic sleep deprivation-induced cognitive impairment, potentially by alleviating oxidative stress and mitochondrial impairment in microglia. NR supplementation also increased REM sleep and reduced NREM sleep by approximately 17%. In human studies, NR improved sleep efficiency in young and middle-aged male individuals with insomnia. It also improved sleep quality and reduced fatigue and drowsiness in older adults. More research is warranted to understand the impacts of NR on sleep for women. **Conclusions**: NR supplementation is a reliable and effective alternative to boost NAD^+^ levels and may ameliorate sleep patterns.

## 1. Introduction

NAD^+^ is a fundamental molecule participating as a redox cofactor in several metabolic reactions [1]. NAD^+^ is used as a substrate in regulation pathways for energetic, genotoxic, and infectious stress [1]. Notably, NAD^+^’s neuroprotective role is associated with oxidative stress.

Despite its critical role in metabolic reactions, NAD^+^ sharply declines with age. Additionally, evidence suggests a negative correlation between NAD^+^ levels and age in human tissues, such as skin, blood, and brain [2], contributing to the pathogenesis of aging-related diseases [3]. Strikingly, NAD^+^ depletion is also observed in neurodegenerative disorders, with NAD^+^ expression linked to beta-amyloid (Aβ) toxicity in Alzheimer’s disease [4].

NAD^+^ depletion negatively impacts sleep and circadian rhythms. A complete sleep cycle consists of two primary types: non-rapid eye movement (NREM) and rapid eye movement (REM) sleep. NREM sleep encompasses three stages: Stage 1 is a transitional stage with theta brain waves. Stage 2 is a light sleep stage characterized by reduced heart-rate variability, slower respiratory rates, and high-frequency brain activity. Stage 3, or slow-wave sleep, is characterized by deep sleep [5]. REM sleep is a deep sleep state characterized by low muscle tone coupled with wake-like brain activity [6]. Circadian rhythms are regulated by an intrinsic central clock, the suprachiasmatic nuclei, driving behavioral and physiological functions in a nearly 24 h cycle [7]. Sleep–wake cycles are governed by a Two-Process Model, where sleep homeostasis and circadian rhythmicity interact harmoniously to promote sleep [8]. Circadian rhythmicity is influenced by zeitgebers, social and environmental time-givers that aid in circadian timing [9]. Light is the most powerful zeitgeber; however, exercising, eating, and other behaviors are other important zeitgebers [9]. In research, Zeitgeber Time (ZT) is standardized to denote a 24 h circadian cycle, where ZT0 refers to the beginning of the light phase and ZT12 refers to the beginning of the dark phase [10].

NAD^+^ is an enzyme cofactor and substrate for the circadian rhythm, interacting with clock gene components, particularly modulating the activity of sirtuins and other NAD-dependent enzymes [11,12]. The sleep–wake cycle and circadian rhythms are disturbed by an age-associated decline in NAD^+^ levels. A potential explanation for this phenomenon is the intrinsic association between sleep, circadian clock rhythmicity, and metabolic homeostasis [13]. Although the vertebrate mammalian circadian system is hierarchically organized with a master hypothalamic clock, the suprachiasmatic nuclei (SCN), primarily responsive to light/dark cycles, the feedback loops from nutrients such as NAD^+^ are essential for molecular clock synchronization in peripheral tissues [13]. NAD-dependent enzymes such as Sirtuin 1 (SIRT1) are associated with and modulate the clock gene expression of CLOCK and BMAL1 [11,14,15]. Thus, the core clock genes’ transcription–translation feedback loop is directly influenced by NAD^+^ levels [15,16,17]. Previous studies demonstrated a potential therapeutic benefit of boosting NAD^+^ to counteract the age-related decline [18]. Therefore, increasing NAD^+^ availability may have therapeutic benefits for sleep disorders and neurodegeneration, while also promoting healthy aging [19,20]. Figure 1 summarizes how NAD^+^ influences molecular clock components.

Dietary supplementation with NAD^+^ precursors significantly attenuates the depletion effects in cognitive deterioration [4], clock gene expression [20], and circadian function [16]. Nicotinamide riboside (NR), also known as a form of vitamin B3, is a biochemical precursor of NAD^+^ [21]. Growing evidence suggests that NR supplementation enhances NAD^+^ bioavailability and prevents the detrimental effects of age-related depletion, including cognitive function, mitochondrial function, and insulin sensitivity [19,22].

The safety and effects of NR administration have been described in the literature in clinical and pre-clinical studies. In rodents, safety for oral, intragastric, and intramuscular NR administration has been established with doses ranging from 100 to 1200 mg/kg/day [2]. In humans, NR and NAD^+^ metabolites are considered safe at 100, 300, and 1000 mg/day doses for adults [23]. Common adverse events and side effects include gastrointestinal symptoms (i.e., stomachache and nausea), fatigue, muscle pain, and skin rashes [23]. Importantly, a trial examining NR safety and metabolic implications of long-term NR administration did not find significant differences in adverse events between placebo and NR groups [24].

Despite the above-described metabolic benefits of NR and NAD^+^ supplementation, few studies focus on the effects of NR administration on sleep patterns, such as duration, quality, fragmentation, and architecture, or sleep stages. This narrative review summarizes the current scientific knowledge on the effects of nicotinamide riboside supplementation on sleep and circadian rhythms.

## 2. Materials and Methods

A narrative review was conducted in April–May 2025 using the PubMed, CINAHL, and Scopus databases, using medical subject headings (MeSH) terms. Emerging articles were exported into an EndNote file (EndNote 2.0 version 20.5.0.18631), and duplicates were removed. Table 1 summarizes the search strategy utilizing MeSH terms and Boolean logic.

### 2.1. Inclusion and Exclusion Criteria

The included studies were peer-reviewed clinical and pre-clinical studies published in English. They focused on dietary supplementation with nicotinamide riboside (NR) metabolites, with an expected outcome of sleep and sleep-related disturbances. Due to the limited scope of results, pre-clinical and clinical studies were included in the final sample. Studies that did not focus on NR or sleep were excluded.

### 2.2. Data Extraction

Data was exported to a citation manager. The articles were read fully after removing duplicates and employing the exclusion criteria. The results from the selected studies were summarized, including overall study characteristics (authors and year of publication), population for clinical studies (age, gender, and sample size), and pre-clinical studies (animal model and experiment description). Regarding dietary supplementation with vitamin B3, the data extracted included dose, duration, forms of NR metabolite selected, administration route, and information on side effects or adverse events. When reported, data from studies were considered significant when *p* < 0.05.

## 3. Results

### 3.1. Literature Search and Studies Characteristics

Eighty-eight studies were identified using the MeSH terms described in Table 1. Eleven articles were included in the final results. Figure 2 illustrates the flowchart of the selected articles.

Findings were allocated into four categories: (1) safety of nicotinamide riboside supplementation, (2) impacts of NR on circadian rhythms and genetic mechanisms, (3) NR improves sleep in animal models, and (4) NR ameliorates sleep patterns in humans. Table 2 summarizes the evidence, distinguishing between clinical and human studies.

### 3.2. Safety of Nicotinamide Riboside Supplementation

Nicotinamide riboside (NR), also known as a form of vitamin B3, is a biochemical precursor of NAD^+^ [21] and is considered better for elevating NAD^+^ levels when compared to other metabolites [2]. NR supplementation is well-tolerated, and it can recover mitochondrial function and augment oxidative phosphorylation, inhibiting microglial pro-inflammatory responses and protecting brain function, potentially by inhibiting neuroinflammation [26].

NR and other NAD^+^ metabolites are considered safe for oral administration in humans and mice and for intramuscular and intragastric administration in animal models. Rodent studies exploring oral (400 mg/kg/day) and intramuscular (1000 mg/kg/day) administration demonstrated a boost in NAD^+^ and superior hepatic concentrations [27]. Intragastric NR supplementation has been implemented in mice, targeting microglia and inflammatory glial cells in response to sleep deprivation [26].

In humans, NR and NAD^+^ metabolites are considered safe for adults at doses of 100, 300, and 1000 mg per day. NR safety and tolerability range from 100, 300, 1000 mg to 2000 mg daily for 12 weeks in middle-aged and older adults [28]. Blood NAD^+^ levels may increase 2.7-fold with a single oral NR dose [27]. Additionally, single doses of 100, 300, and 1000 mg of NR boost the NAD^+^ metabolome in humans [27].

Irie and colleagues [19] investigated the pharmacokinetics and safety of oral administration of nicotinamide metabolites (nicotinamide mononucleotide (NMN)) in humans. Their non-blinded clinical trial enrolled 10 healthy males who received 100, 250, or 500 mg of nicotinamide in capsules at 9 AM after overnight fasting at each visit, followed for five hours at rest with drinking only water. Urine collections were conducted every 30 to 60 min for the first two hours and at the end of the study. In addition, blood samples were collected every 5 to 20 min for the first 1 h, followed by every 30 to 120 min for the next four hours. The results indicate that oral NMN did not cause significant clinical symptoms or changes in heart rate, blood pressure, oxygen saturation, and body temperature [19]. Participants also had unremarkable laboratory results, aside from increases in serum bilirubin levels and decreases in serum creatinine, chloride, and blood glucose levels, all within normal ranges, regardless of the NMN dose. No side effects or adverse events were reported with oral vitamin B3 supplementation in older adults [29].

Common adverse events and side effects include gastrointestinal symptoms (i.e., stomachache and nausea). Importantly, a trial examining NR safety and metabolic implications of NR long-term administration did not find significant differences in adverse events between the placebo and NR groups [24]. Similarly, other studies with high-dose NR also reported no serious adverse events [27].

### 3.3. Impacts of NR Supplementation on Circadian Rhythms and Genetic Mechanisms

NAD^+^ plays a crucial role in circadian clock function and metabolic homeostasis, including sleep–wake and circadian disruption, which are hallmarks of aging. NAD-dependent sirtuins, particularly SIRT1, are essential to regulate metabolism and circadian rhythms [30]. Genetic disruption of the circadian clock leads to obesity, metabolic disorders, and sleep disturbances [29].

Levine and colleagues hypothesize that reversing the decline in NAD^+^ would restore clock activity [16]. They examined pharmacologic and genetic manipulations of NAD^+^ in the control of genome-wide circadian transcriptional rhythms and identified NAD^+^ as a mediator of PER2 nuclear extrusion and BMAL1 chromatin binding activity. Four-month-old mice were supplemented with 400 mg/kg body weight/day NR in drinking water. Animals had access to NR ad libitum for 4 months and were housed under a 12:12 light/dark cycle. Analysis indicated that NR supplementation increased NAD^+^ levels 4–5-fold in the liver, soleus muscle, and hypothalamus. The authors also demonstrated that NR supplementation enhanced BMAL1 chromatin binding genome-wide [16]. Lastly, NR supplementation in 8-month-old mice also restored NAD^+^ to equivalent youth levels [16].

### 3.4. NR Supplementation Improves Sleep in Animal Models

The natural age-related decline in NAD^+^ is linked to sleep–wake disturbances [16]. In addition, decreases in NAD^+^ levels negatively impact sleep quality [29]. Thus, supplementation with dietary precursors could re-establish NAD^+^ biosynthesis and increase intracellular levels, potentially reversing physiological decline and preventing diseases.

Beaton and colleagues examined the sleep–wake cycle of random-bred Swiss mice and rats treated with L-methionine plus L-histidine, L-methionine plus nicotinamide, L-histidine, and nicotinamide [31]. Treatment included daily injections of 250 mg/kg of compounds administered for 21 consecutive days. The results suggested that nicotinamide produced a significant increase in rapid eye movement (REM) sleep [25,31]. Additionally, rats injected with nicotinamide expressed an increase in serotonin in the brain, which may have impacted their sleep [32].

Interestingly, Bushana and colleagues [21] found a reduction in non-rapid eye movement (NREM) sleep in mice supplemented with NR. The authors examined sleep need and duration in 33 C57BL/6J mice (12 females and 11 males aged 10–12 weeks) singly housed in a 12:12 light–dark cycle. After isoflurane anesthesia, mice were implanted with EEG and electromyographic (EMG) electrodes, received post-operative analgesia, and recovered for 12–13 days. At 10 weeks of age, mice were randomized into two groups: control chow or NR (3.33 g/kg) for 10 weeks, resulting in a daily NR consumption of 400 mg/kg. Mice were given food and water ad libitum; the diet and animals were weighed twice weekly [21]. After recovery, baseline EMG recordings began with 24 h of undisturbed monitoring, starting at ZT1. Sleep deprivation consisted of a 6 h gentle handling during protocol, from ZT1 to ZT7, which encouraged quiet wakefulness during 30 min recovery sleep episodes, increasing opportunities to study the effect of NR on the waking EEG.

Bushana and colleagues described that long-term (6–10 weeks) dietary supplementation with NR reduced the time that mice spent in NREM sleep by 17% and accelerated the rate of discharge of sleep need according to a mathematical model of sleep homeostasis [21]. The authors concluded that increasing nicotinamide availability reduces sleep need and improves the cortical capacity for energetically demanding high-frequency oscillations [21].

Considering that neuroinflammation plays a key role in sleep deprivation-induced memory loss and cognitive impairment, NAD^+^’s anti-inflammatory role and redox reaction may lead to mitochondria-protective effects in microglia, reduce inflammation, and ameliorate cognitive impairment [26]. Recently, Chen and colleagues demonstrated that NR supplementation mitigated cognitive and memory impairment in sleep-deprived mice [26]. The authors randomized male C57BL/6 mice (6–8 weeks, 22–26 g) into control, chronic sleep restriction (CSR), NAD^+^, and NAD^+^/CSR groups. They performed a 14-day sleep restriction protocol with 18 h of forced locomotion using a slowly rotating stick. The mice received intragastrical NAD^+^ supplementation with 250 mg/kg/day.

NAD^+^ supplementation mitigated CSR-induced cognitive decline and memory impairment, with mice exhibiting a lower level of avoidance in an open field test [26]. Similarly, NAD^+^ treatment mitigated spatial memory deficits and spontaneous activity in the Morris Water Maze [26]. Additionally, the in vitro results demonstrated that NAD^+^ inhibited CSR-induced microglial activation and alleviated CSR-mediated oxidative stress and mitochondrial impairment in microglia [26].

### 3.5. NR Ameliorates Sleep Patterns in Humans

NAD^+^ and NR may recover sleep as a consequence of metabolic improvement. NAD^+^ is an obligatory cofactor for oxidation/reduction and is also involved in glucose metabolism, which is associated with NREM sleep [21]. An association between poor glucose control, shorter sleep duration, and poorer sleep quality can also indicate a bidirectional relationship between sleep and metabolism [33]. Lengthy sleep onset and fragmentation are associated with increased insulin resistance and glucose intolerance. Additionally, glycemic variation may reflect sleep fragmentation, exhibiting a negative correlation with total sleep duration and a positive correlation with wake after sleep onset and sleep onset latency [33]. Furthermore, the brain accumulates oxidative stress during wakefulness and eliminates biomarkers of oxidative stress during sleep. Lastly, the downregulation of glucose metabolism is central to the sleep homeostatic process [21].

Robison and colleagues compared the outcomes of NR administration in people with normal sleep and those with moderate to severe insomnia, suggesting a role of the amino acid tryptophan, which can be converted into serotonin (5-HT) and niacin [34]. Participants received NR supplementation twice daily for 23 days, with a progressively increased dose from 1 g on the first day to 3 g from day 3 to 23. In-lab sleep recordings were performed on nights 14–16 and 21–23 during NR supplementation and repeated on nights 19–21 after supplementation stopped. Sleep recordings included an electroencephalogram (EEG), an electromyogram (EMG), and an electrooculogram (EOG). Despite the limitations, such as small sample (six with normal sleep, three males and two females; and two females with severe insomnia) and age inconsistency (21–74 for normal sleep and 40–50 for insomnia), the findings revealed a significant increase in REM sleep and improvement in sleep efficiency for all participants with insomnia [34].

Morifuji and team evaluated the impacts of a 12-week supplementation with 250 mg/day β-nicotinamide mononucleotide on sleep and physical function [35]. Participants were sixty Japanese male and female older adults aged 65–75. Fasting blood and urine NAD analytes, physical performance with a walking test, and sleep quality with the Pittsburgh Sleep Quality Index were assessed at baseline and weeks 4 and 12. Participants in the supplement group exhibited a shorter 4 min walk test, improved sleep quality, and higher blood levels of NAD^+^ and its metabolites at 12 weeks [35].

Similarly, a 12-week supplementation with vitamin B3 improved sleep quality, drowsiness, and fatigue in older adults [29]. Kim and colleagues randomized 108 Japanese older adults (mean age, 72.2 years) into a double-blinded trial with 250 mg of vitamin B3 (as NMN) or a placebo for 12 weeks. Sleep quality was assessed with the Pittsburgh Sleep Quality Index [29]. The authors found a significant main effect of time on sleep duration, sleep disturbance score, daytime dysfunction score, sleep quality score, and total PSQI global score in participants supplemented with vitamin B3 [29].

Interestingly, a cohort study utilizing data from 2084 women from the Korea National Health and Nutrition Examination Survey (KNHASES) 2016–2018 examined the relationship between nutrient intake, comorbid conditions, and habitual sleep in pre- and postmenopausal women [36]. Dietary supplementation included total vitamin A, retinol, B vitamins (B1, B2, and B3), C, folic acid, and minerals (sodium, calcium, iron, phosphorus, and potassium). Habitual sleep duration was classified as very short (<5 h per night), short (5–6 h per night), normal (7–8 h per night), and long (≥9 h per night). Higher vitamin B1, B3, vitamin C, iron, potassium, phosphorus, and calcium intake was negatively associated with short sleep duration in premenopausal women. Their reported mean vitamin B3 intake was 11.83 mg (4.26–25.85) compared to 9.31 mg (3.48–19.55) in postmenopausal women. Further research is warranted, particularly with targeted vitamin B3, female participants, and perimenopause assessments. Figure 3 presents a graphical summary of the current evidence on the effects of vitamin B3 on sleep patterns and circadian rhythms in clinical and pre-clinical studies.

## 4. Conclusions

Nicotinamide riboside is considered safe for use in clinical and pre-clinical studies. NR supplementation replenished the age-related decline in NAD^+^ and restored its youthful levels in mice. Pre-clinical studies suggest that NR enhances the performance of the clock genes BMAL1 and PER2, ameliorates chronic sleep deprivation-induced cognitive impairment, and may alleviate oxidative stress and mitochondrial impairment in microglia. NR supplementation also increased REM sleep and reduced NREM sleep by approximately 17%. In humans, NR improved sleep efficiency in young and middle-aged male individuals with insomnia. It also improved sleep quality and reduced fatigue and drowsiness in older adults. More research is warranted to understand the impacts of NR on sleep for perimenopausal women.

Despite limitations such as the limited number of clinical trials, the lack of diversity in the studies, and the use of NR supplementation in combination with other nutrients, this narrative review provides an overview of the current state of science of a safe, reliable, and effective alternative to boost NAD^+^ levels. This may benefit the age-related decline in NAD^+^ and ultimately ameliorate sleep quality, duration, and circadian clock function. Although promising, these findings require further support from clinical studies that target women, diverse demographic backgrounds, and a broader range of age groups, incorporating both objective and subjective sleep assessments.

## Figures and Tables

**Figure 1 nutrients-17-01982-f001:**
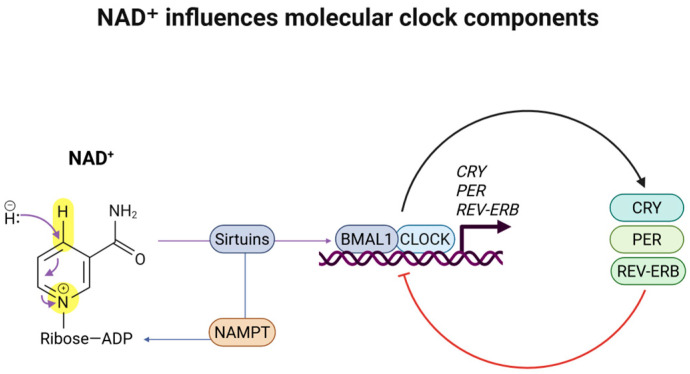
NAD^+^ influences molecular clock components. NAD^+^ is a cofactor for SIRT1, a critical protein deacetylase for circadian clock regulation. SIRT1 is recruited to the CLOCK: BMAL1 chromatin complex, regulating the amplitude and duration of circadian gene expression.

**Figure 2 nutrients-17-01982-f002:**
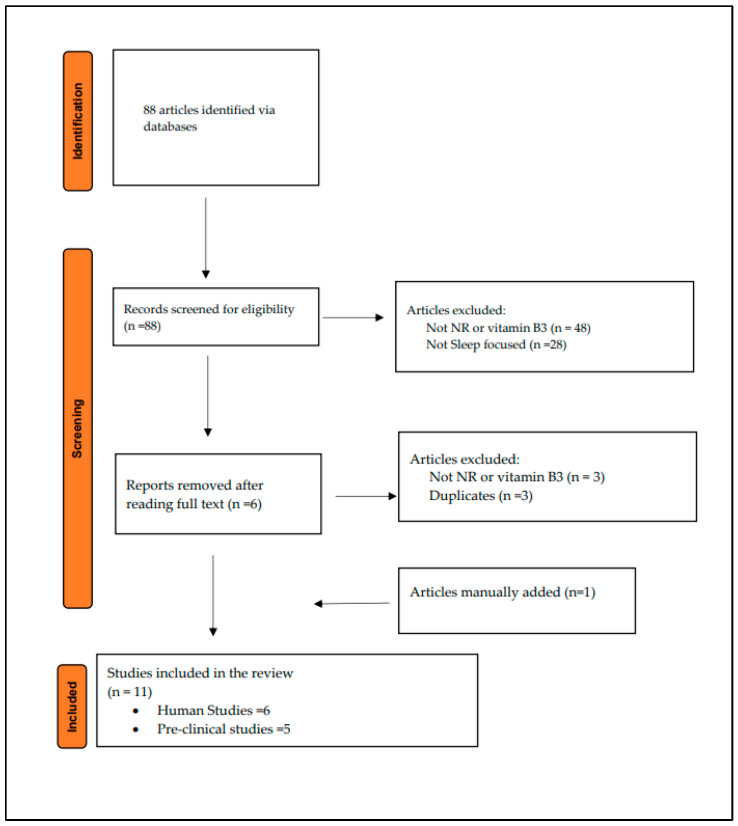
Literature search flowchart.

**Figure 3 nutrients-17-01982-f003:**
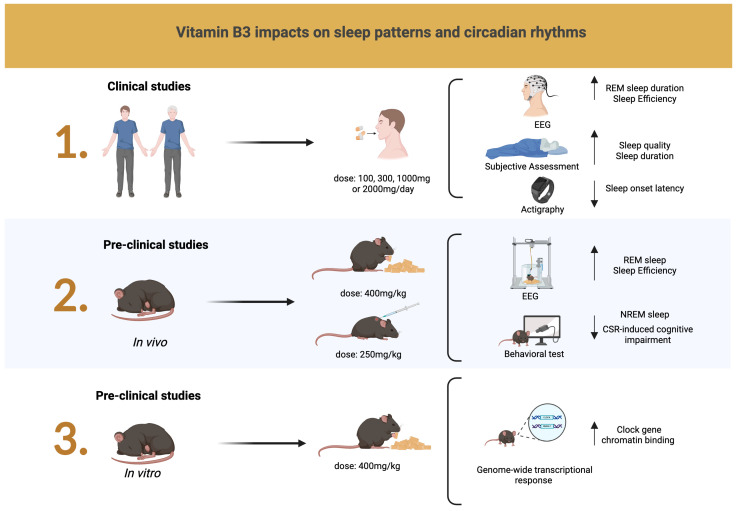
In clinical studies, an EEG demonstrated an increase in REM sleep duration and sleep efficiency, while subjective assessments with the Pittsburgh Sleep Quality Index and the Karolinska Sleepiness Scale indicated improvements in sleep quality and duration. Actigraphy findings indicated reduced sleep onset. Pre-clinical studies reported improvements in REM sleep, increased sleep efficiency, reduced NREM sleep, and mitigated cognitive impairment induced by CSR. Arrows pointing up indicate an increase. Arrows pointing down indicate a decrease. Created in https://BioRender.com.

**Table 1 nutrients-17-01982-t001:** Medical subject heading (MeSH) terms.

MeSH Terms	Search Results
((nicotinamide riboside) OR (vitamin B3)) AND (sleep)	87
((nicotinamide riboside) AND (vitamin B3)) AND (sleep)	1

**Table 2 nutrients-17-01982-t002:** Summary of evidence. Arrows pointing UP indicate an increase. Arrows pointing DOWN indicate a decrease.

Author	Year	Study Population	NR Supplementation Regimen	Treatment Type and Duration	Sleep Assessments	Sleep Findings
**Pre-Clinical Studies**
Beaton, et al. [25]	1974	Mouse	250 mg/kg nicotinamide riboside	Daily injections	Sleep–wake cycle	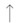 REM (mice)
Beaton, et al. [19]	1975	Swiss mice and rats	250 mg/kg. L-methionine plus L-histidine, L-methionine plus nicotinamide, L-histidine, and nicotinamideControl: L-methionine plus L-serine, L-cysteine, L-serine, and saline	Daily injections 21 days	Sleep–wake cycle	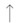 REM (mice)
Bushana et al. [12]	2023	C57BL/6J mice on a 12:12 LD cyclen = 33, 12 females, 11 males; aged 10–12 weeks	Diet containing 3.33 g NR/kg	6–10 weeks	Implanted with an EEG and electromyographic (EMG) to monitor a 6 h sleep deprivation protocol and 30 min recovery sessions	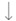 17% NREMS 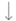 Cumulative manifestation of sleep need during quiet wake
Chen et al. [14]	2024	C57BL/6 mice (6–8 weeks, 22–26 g)	Intragastrical nicotinamide adenine dinucleotide administered at 250 mg/kg/day	14 days	Chronic sleep restriction (CSR) with a forced locomotion protocol	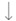 CSR-induced cognitive decline
Levine et al. [8]	2020	4-month-old mice housed in a 12:12 LD cycle	400 mg/kg/day of NR	Four months	Clock gene PER2 and BMAL1 expression	Restored BMAL1 levelsEnhanced PER2
**Human Studies**
Irie et al. [9]	2020	10 healthy Japanese males.	100, 250, and 500 mg of nicotinamide mononucleotide (NMN) for a 5 h pharmacokinetics assessment	Single oral administration	Pittsburgh Sleep Quality Index (PSQI)	No significant findings
Kim et al. [17]	2022	108 Japanese older adults(mean age 72.2)	250 mg of vitamin B3 or placebo for 12 weeks	Daily oral administration	Pittsburgh Sleep Quality Index (PSQI)	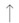 Sleep duration 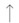 Sleep quality (PSQI global score)
Morifuji et al. [23]	2024	60 Japanese older adults(mean age 69.0)	250 mg of β-nicotinamide mononucleotide or placebo for 12 weeks	Daily oral administration	Pittsburgh Sleep Quality Index (PSQI	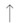 Sleep quality (PSQI global score)
Nguyen et al. [26]	2023	2084 pre- and postmenopausal Korean women 18–80 years old (KNHASES Study, 2016–2018)	12 nutrients (vitamin B1, B3, vitamin C, PUFA, *n*-6 fatty acid, iron, potassium, phosphorus, calcium, fiber, carbohydrate)	Daily oral administration	Habitual sleep duration per night: <5 h = very short 5–6 = short 7–8 h = normal≥9 = long	
Robinson, et al. [22]	1977	6 adults with normal sleep (3 males and 2 females) and 2 females with severe insomnia	1–3 g of NR twice daily for 23 days	Daily oral administration	In-lab sleep recording	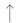 Sleep efficiency 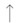 REM sleep duration
Soon et al. [24]	2025	43 adults	Randomized, crossover, double-blind, controlled trialVitamin B3 (1.93 mg) in addition to mulberry leaf extract (750 mg), whey protein containing 120 mg tryptophan, zinc (1.35 mg), magnesium (12.6 mg), and B6 (0.135 mg)Control (4 g wheat protein hydrolysate)	Morning vs. evening 14-day oral administration + 28-day washout period	ActigraphyKarolinska Sleepiness Scale	Evening supplements: 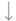 Sleep onset 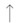 Sleep quality

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
