# Peer review of "Vitamin B3 Ameliorates Sleep Duration and Quality in Clinical and Pre-Clinical Studies"

_nutrients, 2025, doi:10.3390/nu17121982_

Round 1
Reviewer 1 Report
Comments and Suggestions for Authors
The aim of the article was to summarize the state of knowledge on the effects of nicotinamide riboside (vitamin B3) supplementation on sleep. The author points out the lack of a synthesis of research on this topic to date, hence the possible underestimation of the issue despite the frequent occurrence of sleep disorders in the elderly, especially in the course of neurodegenerative processes such as Alzheimer's disease and other dementias. I believe that the question posed by the author is original and well-defined and that the article falls within the scope of Nutriens' interests.
The review has been prepared in accordance with the principles adopted for this type of studies. It includes 10 articles devoted to the following issues: (1) safety of nicotinamide riboside supplementation (2) effects of NR on circadian rhythms and genetic mechanisms, (3) NR improves sleep in animal models, and (4) NR ameliorates sleep patterns in humans. These articles were discussed factually, and the conclusions formulated by the authors were properly interpreted.
The author concludes that despite some limitations, such as the small number of clinical trials, the lack of diversity in the studies, and the use of NR supplementation in combination with other nutrients, her review suggests that NR supplementation may be considered a safe, reliable, and effective alternative to prevent the age-related decline in NAD+ levels, which is crucial for sleep and neurodegeneration.
Author Response
Thank you for the reviewer's comments. The updated version submitted highlights the edits in yellow.
Sincerely,
Carleara Weiss, PhD.
Reviewer 2 Report
Comments and Suggestions for Authors
Summary
This study sought to review literature pertaining to nicotinamide riboside supplementation effects on sleep outcomes. The paper is interesting and may add to the literature base, yet various amendments need to be made prior to being considered for publication.
Abstract
Line 12: there may be better word selection to finish this sentence rather than “may replenish this burden”
Line 14: abbreviation has been defined twice
Line 19: the author alternates between reference to animal and human research findings, yet this could be articulated better. I.e. when discussing findings from either pre-clinical or clinical studies, articulate this within your statements
Line 26: given the limited literature base, this statement should be more broad than emphasising premenopausal women
Line 26-27: this conclusion needs to be amended to improve validity. First the certainty of NR supplementation is overstated. Next, claiming NR “prevents” NAD⁺ decline implies long-term, population-wide preventative efficacy. NR may increase NAD⁺ levels acutely, but whether it prevents age-related decline long-term or meaningfully alters health trajectories remains uncertain.
Also, the statement that NR has “significant benefits for sleep” is not robustly supported by current evidence.
Introduction
Line 38: classification of “strong” correlation varies, please be more specific
Line 45: should have citation to support
Line 57: it would be useful for the author to discuss this in more descriptive detail. What type of studies were these findings derived from?
The introduction is quite brief. There is room to expand discussion of literature on the relationship between NR, NAD+ and sleep
Methods
Line 70: I think this results heading here is an error and needs to be deleted
Line 86: do you mean p<0.05?
Results/Discussion
Line 118: The paragraph currently reads as underdeveloped, with limited connection to the subsection’s core argument. Further detail or contextualisation would strengthen its contribution
Line 229: an additional paragraph should be added, discussing collective limitations of the literature base and also recommendations for future research, as the current literature base on this subject is not strong
Literature table: in the author column, also add the number reference as it appears in the final reference list
Conclusions
Line 239: acknowledgement of literature limitations is understated, this could be expanded on
Line 244: The conclusion asserts outcomes with a degree of confidence that may not be warranted given the preliminary or limited nature of the data. Moderating the claims would strengthen the scientific rigour
Line 261: the volume of literature utilised is limited. A more robust discussion in the introduction may enhance.
Author Response
Dear Reviewer 2,
Thank you for your precious suggestions. I have attached the responses and indications of changes in the updated version.
Sincerely,
Carleara Weiss, PhD

Reviewer 3 Report
Comments and Suggestions for Authors
Nicotinamide adenine dinucleotide (NAD⁺) is a critical coenzyme involved in cellular metabolism, ATP production, and DNA repair. Beyond its metabolic roles, NAD⁺ plays a fundamental part in modulating circadian rhythms and sleep homeostasis. Its regulatory functions are mediated through several mechanisms, including SIRT1-dependent repression of circadian gene expression and epigenetic modifications. Furthermore, NAD⁺ levels exhibit circadian oscillations, linking cellular metabolism with the molecular clock. Given its roles in both aging and sleep physiology, summarizing recent progress in NAD⁺ research provides not only mechanistic insights but also opens avenues for therapeutic intervention in sleep and circadian disorders.
Overall, this review is well-structured and timely. However, the following changes are required to improve its clarity and comprehensiveness prior to publication:
-
The introduction would benefit from a more thorough explanation of circadian biology and sleep stages. Specifically, the distinctions between REM (Rapid Eye Movement) and NREM (Non-Rapid Eye Movement) sleep should be clarified. Additionally, the concept of Zeitgeber Time (ZT)—which denotes time based on environmental cues like light-dark cycles—should be introduced, as it is fundamental in circadian research.
-
Including a schematic diagram summarizing how NAD⁺ influences molecular clock components (e.g., SIRT1, CLOCK/BMAL1, PARP, and AMPK) would enhance the accessibility of the review for a broader scientific audience. Such a visual could clarify the complex interplay between NAD⁺ metabolism and circadian gene regulation.
-
Additional references should be included:
-
Terzi et al., “Phylogenetic conservation of the interdependent homeostatic relationship of sleep regulation and redox metabolism.”
-
Davinelli et al., “Sleep and Oxidative Stress: Current Perspectives on the Role of NRF2.”
-
Nakahata et al., “Circadian Control of the NAD⁺ Salvage Pathway by CLOCK-SIRT1.”
-
Chang et al., “SIRT1 Mediates Central Circadian Control in the SCN by a Mechanism that Decays with Aging.”
-
Peek et al., “Circadian Clock NAD⁺ Cycle Drives Mitochondrial Oxidative Metabolism in Mice.”
-
Rajman et al., “Therapeutic Potential of NAD⁺-Boosting Molecules: The In Vivo Evidence.”
-
Author Response
Dear Reviewer 3,
Thank you for the suggestions. The updated version reflects the edits and recommendations.
I included two figures based on your recommendation, and I would appreciate your feedback on them.
Sincerely,
Carleara Weiss, PhD.

Round 2
Reviewer 2 Report
Comments and Suggestions for Authors
The author responses have addressed my initial feedback, thanks
Reviewer 3 Report
Comments and Suggestions for Authors
I'm happy to see my comments/questions have been fully addressed and have no further questions prior to its publication.